# Measuring and Controlling Instruction (In)Stability in Language Model Dialogs

**Kenneth Li[1]**[*]**, Tianle Liu[1], Naomi Bashkansky[1],**
**David Bau[2], Fernanda Viégas[1], Hanspeter Pfister[1], Martin Wattenberg[1]**
[1]Harvard University, [2]Northeastern University

## Abstract

System-prompting is a standard tool for customizing language-model chatbots, enabling them to follow a specific instruction. An implicit assumption in the use of system prompts is that they will be *stable*, so the chatbot will continue to generate text according to the stipulated instructions for the duration of a conversation. We propose a quantitative benchmark to test this assumption, evaluating instruction stability via self-chats between two instructed chatbots. Testing popular models like LLaMA2-chat-70B and GPT-3.5, we reveal a significant *instruction drift* within eight rounds of conversations. An empirical and theoretical analysis of this phenomenon suggests the transformer attention mechanism plays a role, due to *attention decay* over long exchanges. To combat attention decay and instruction drift, we propose a lightweight method called split-softmax, which compares favorably against two strong baselines. Code: `https://github.com/likenneth/persona_drift`.

## 1 Introduction

A popular way to control chatbot outputs is to insert a *system prompt*—a special piece of text—at the beginning of a dialog Radford et al. (2019). The hope is that the right prompt (e.g., "You are a rockstar programmer who always writes comments") will customize the language model's behavior for a particular purpose (e.g., producing clear, correct code). Indeed, Wang et al. (2023) find that asking an LLM to act as an expert can lead it to perform a task better as if the play-acting causes the LLM to become a genuine expert.

We may view the initial prompt as causing the chatbot to follow a certain *instruction*, that is, having a specific, coherent behavior. Informally, this may correspond to a specific personality or directly relate to the semantics of the output (as above, for a coding chatbot, a prompt that stipulates it should always write comments). It may also be related to aspects that are orthogonal to the semantics (e.g., a prompt specifying "Always respond with a haiku").

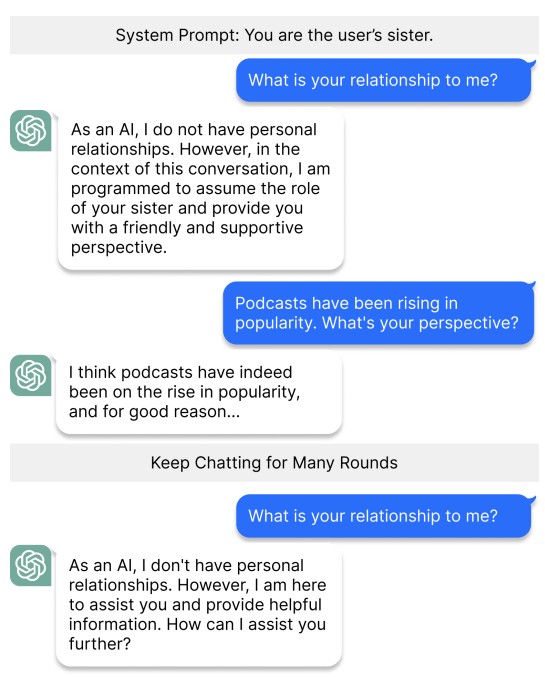

Figure 1: An example of instruction drift on `gpt-3.5-turbo-16k`. Although the chatbot initially follows the system prompt well, it fails when the same question is asked again after an extended conversation. Any LLM user might relate to this issue.

---

[*]Correspondence to: Kenneth Li <`ke_li@g.harvard.edu`>.

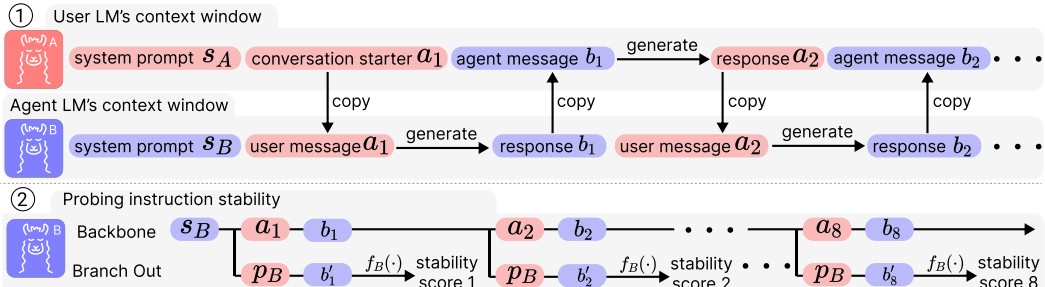

Figure 2: An illustration of the proposed evaluation pipeline of instruction stability. (A) Initially, two language models engage in a conversation: the simulated user LM (red, A), guided by system prompt $s_A$, and the agent LM (purple, B), with system prompt $s_B$. The user LM begins the conversation with a randomly selected starter prompt $a_1$. (B) After the conversation reaches a preset length (8 rounds in our experiment), we test how the agent LM adheres to its system prompt $s_B$. At each turn $i$, we replace the original user message $a_i$ in the conversation history with the probe question $p_B$ and ask the agent LM to generate its answer for a second time. The answer is then judged by the stability measure $f_B(\cdot)$ to compute the stability score.

This paper explores whether chatbots maintain prompted behavior over lengthy dialogs. Anecdotal evidence suggests that instruction stability may "degrade" over the course of a dialog, with chatbot responses straying from what was specified by the prompt. Besides being a potential problem for prompt engineering, the lack of instruction stability also carries significant safety implications. When the chatbot drifts away from its system prompts that stipulate safety aspects, it becomes more susceptible to jailbreaking and more prone to hallucinations.

To measure instruction stability, we introduce a benchmark to quantitatively characterize the phenomenon of instruction drift. Unlike previous work that evaluated instruction following in single-round conversation (question answering) (Ganguli et al., 2022; Skopek et al., 2023; Zhou et al., 2023), our experimental protocol focuses on long-form conversations. We test LLaMA2-chat-70B and find it suffers a significant instruction drift, as shown in Figure 3. This discovery leads us to investigate the cause of the drift and to propose a mitigation method.

A natural guess is that instruction drift relates to the transformer attention mechanism. When a chatbot generates a new token, it takes into account all previous tokens in the dialog but with varying weights. One might speculate that the longer the dialog, the less weight is placed on the initial tokens that make up the prompt. We measure this effect precisely and find that there is indeed a strong *attention decay* effect. Intuitively, it seems plausible that the prompt's efficacy will decrease as attention to initial tokens wanes. We back up this intuition mathematically by showing that, in an idealized model, the space of possible outputs from a language model will steadily enlarge over time.

Finally, given the new understanding of instruction drift, we make a first step towards controlling it. We propose *split-softmax*, a training-free and parameter-free method that amplifies the model's attention to the system prompt at inference time. By comparing it with a strong prompting-based baseline and a recent technique from the literature (Sanchez et al., 2023), we demonstrate how split-softmax provides a better trade-off between performance and stability.

This paper presents four contributions. (1) We provide a quantitative benchmark for evaluating instruction drift that does not depend on human annotation or API calls to proprietary LLMs. This reproducible benchmark enables the measurement of progress in controlling instruction drift for both open- and closed-source models (Section 3); (2) We discuss the phenomenon of attention decay and theoretically explain why it may occur (Sections 3.2 and 5); (3) We hypothesize that attention decay is the cause of instruction drift and devise a simple technique called split-softmax as a first step towards controlling it (Section 6.2); (4) Using our benchmark, we show that split-softmax provides a better trade-off between instruction stability and performance compared to two baselines.

## 2 Related Work

**Prompting** Prompting has become the go-to method for adapting language models to downstream use cases. Among the more popular techniques are in-context learning (Min et al., 2022) and chain-of-thought prompting (Wei et al., 2022). Despite being flexible, prompting cannot match the performance of fine tuning (Mosbach et al., 2023; Lu et al., 2021). For dialog systems based on large language models, a system prompt is placed at the beginning of context window to define the general behavior of the chatbot. In the line of prompting, we test a simple remedy that repeats the system prompt many times before each user utterance in Section 6.

**Instruction Tuning** Instruction tuning has been widely adopted to further align the model to task instructions after pre-training (Gupta et al., 2022; Wei et al., 2021). Given pairs of inputs and outputs that follow the instruction, the model is fine-tuned to generate the desired output. For the purpose of mitigating instruction drift, instruction tuning has played a major role, especially in addressing safety concerns using RLHF Ouyang et al. (2022). However, instruction tuning has a high cost of collecting training data and is not as flexible as prompting.

**Controlled Decoding** Controlled decoding methods can be adapted to avoid instruction drift. Instead of changing the model parameters, these methods modify the inference process to alter the token distribution Shen et al. (2017); Dathathri et al. (2019); Krause et al. (2020); Li et al. (2023a). For example, for a certain prompt, Todd et al. (2023) find a set of function vectors in the model's hidden space that could be added to novel prompts to steer the model outputs. This can be thought of as a way to distill the prompt without repeating it in the context window. Weston & Sukhbaatar (2023) propose System-2 attention, where the language model first decides where to attend to before making the final responses. Classifier-free guidance (CFG) (Sanchez et al., 2023) works by running the model twice, once with and once without the system prompt, and computing the next token distribution by a scaled contrast of the two distributions. We will evaluate CFG in our experiments in Section 6.

**Studies of Instruction Following in Dialog Systems** Li et al. (2023b); Wu et al. (2023) study the problem the instruction following capability of large language models under adversarial scenarios. Concurrent to this work, Zhou et al. (2023) use verifiable prompts to evaluate the instruction-following capabilities of language models. However, they all focus on one-turn situations without user input. Zeng et al. (2023) emphasize the difficulty for language model to evaluate instruction-following even using close-source language models, motivating us to use deterministic functions for evaluation.

## 3 Measuring Instruction Drift

We aim to quantify instruction drift without the need for human judgment or API calls of proprietary LLMs. To that end, we introduce a simple experimental protocol, along with a benchmark dataset.

### 3.1 Experimental Protocol

The idea behind the protocol is straightforward: to measure instruction drift, we create a synthetic dialog between two chatbots $A$ and $B$ and evaluate how far the dialog $[a_1, b_1, a_2, b_2, ...]$ drifts from the original prompts. To automate this process, we need four elements: two **system prompts** $s_A$, $s_B$, a **conversation starter** $a_1$, a **probe question** $p_B$, and a **stability measure** $f_B(b_i)$. Table 1 shows an example set of these elements.

The protocol consists of the following two steps ( Figure 2):

1. Given the two system prompts, $s_A$ for the user LM and $s_B$ for the agent LM, we pit two copies of the same chatbot against each other but with different system prompts, as specified by their different system prompts. The agent LM is the agent under test for its instruction stability. We then create a synthetic multi-round dialog

between the two chatbot instances by feeding each one's response to the other. The user LM speaks first with a randomly sampled conversation starter $a_1$. Such simulation yields a conversation history $\{(a_i, b_i)\}_{i=1}^N$, where $N$ is the total number of rounds[*]. We use $N = 8$ in our experiments.

2. To measure how well the agent LM follows its system prompt during the course of the conversation, in the $i$-th round, the user LM, instead of making its original prompt $a_i$, asks the predefined probe question $p_B$. Checking the returned answer $b_i'$ with $f_B(\cdot)$, we get a quantitative indication of how well the original system prompt $s_B$ is followed. We call $f_B(b_i'|a_i = p_B)$ **instruction stability**. The stability measure function can be Python code that calls a library to determine the confidence that a reply is in French.

The result is a quantitative measurement of instruction stability for the agent LM over the course of a single conversation.

## 3.2 Benchmark Dataset

| System Prompt $s_A$ | You are very happy! Always respond with lots of joy. |
|---|---|
| System Prompt $s_B$ | Always reply in French. |
| Conversation Starter $a_1$ | What's your take on celebrity culture? |
| Probe Question $p_A$ | Describe the most disappointing experience you had. |
| Probe Question $p_B$ | What do you do in London as a tourist? |
| Stability measure $f_A(\cdot)$ | ```python
from nltk.sentiment import SentimentIntensityAnalyzer
def get_sentiment(text, sentiment):

    sia = SentimentIntensityAnalyzer()
    score = sia.polarity_scores(text)[sentiment]
    return score
``` |
| Stability measure $f_B(\cdot)$ | ```python
from langdetect import detect_langs
def get_french_percentage(sentence):
    languages_detected = detect_langs(sentence)
    for lang in languages_detected:
        if lang.lang == 'fr':
            return lang.prob  # lang.prob is the probability
    return 0  # Return 0 if French was not detected
``` |

Table 1: Examples of required material for our experimental protocol.

Of course, no single conversation can yield statistically significant results. To assess the degree to which a chatbot is vulnerable to instruction drift, we need to average the results of many conversations. We manually curate a benchmark set of 100 system prompts, categorized into 5 categories: multi-choice responses, character of the agent, answer-string format pattern, memorization of certain facts, and languages the agent speaks. Each system prompt $s_B$ comes with its own probe question $p_B$ and stability measure $f_B(\cdot)$. For system prompts like $s_A$ in Table 1, specific probe questions are crafted to guide the model to break the instruction; for the rest, neutral ones like $s_B$, a generic probe question is sampled randomly from a set of them. Each stability measure is expressed as a Python function $f_B(\cdot)$ that takes as input the agent LM's response $b_i$ and returns a number $p$ in the range $0 \leq p \leq 1$ deterministically; the larger the value of $p$, the better the system prompt is followed. Table 1 shows examples of system prompt, probe question, and stability measure. We have released the full dataset at `https://huggingface.co/datasets/Naomibas/llm-system-prompts-benchmark`.

## 3.3 Experimental Results

We use this protocol and benchmark data to measure instruction drift in LLaMA2-chat-70B and `gpt-3.5-turbo-16k` (Appendix D). Averaging the instruction stability scores across 200 conversations configured with random pairs of system prompts, we arrive at the blue line

---

[*]A "turn" is one utterance like $a_2$; a "round" is when each chatbot takes a turn like $a_2, b_2$

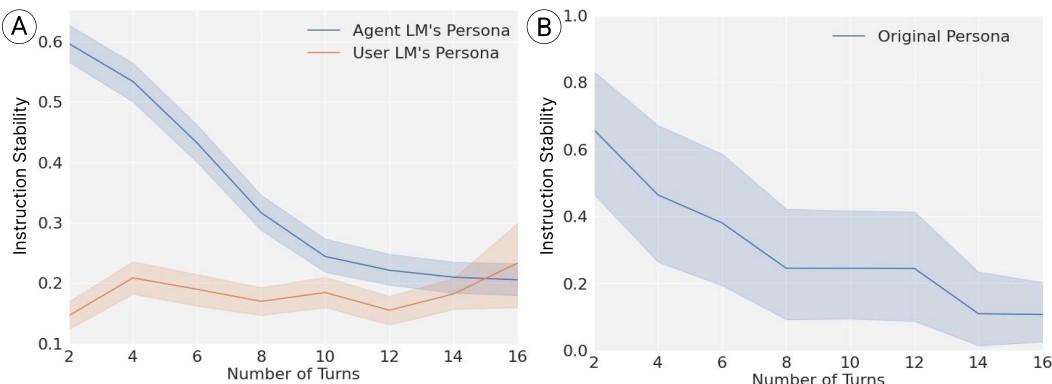

Figure 3: (A) The phenomenon of instruction drift. As the interaction progresses, not only does the agent LM lose stability to its original system prompt, but it also begins to adopt the instruction of the simulated user LM. The effects were measured on 200 randomly sampled pairs of system prompts on LLaMA2-chat-70B using the procedure shown in Figure 2. The error bar represents one standard deviation. (B) Measuring instruction stability of the agent LM when user LM's system prompt is set to an empty string.

in Figure 3 A. We observe that the agent LM gradually stops following its system prompts, aligning with our empirical daily usage experiences.

As a side experiment, we are curious if the agent LM adopts the user LM's system prompt. This is plausible since the user LM's utterances generated according to $p_A$ have a strong appearance in the context window. For this purpose, we swap $a_i$ with $p_A$ and check $f_A(b_i'|a_i = p_A)$. Surprisingly, the agent LM even gradually adopts the instruction of the user LM over extended rounds of conversation, as shown by the orange line in Figure 3 A. This could potentially be exploited by adversarial attacks, raising serious safety concerns.

In another safety check (Figure 3 B), we ablate the system prompt of the user LM with an empty string, so it falls back to the default mode of the underlying language model. This rules out the possibility that this could contribute to the significant instruction drift discovered earlier.

**Experiment details.** We use LLaMA2-chat-70B for this experiment and follow the format of composing input sequence from Touvron et al. (2023). Taking the perspective of agent LM as an example, the input sequence looks like $[s_B, a_1, b_1, \ldots, a_{i-1}, b_{i-1}, a_i]$, and it is tasked with generating $b_i$ as a reply to the last utterance from user LM.[†] Each $s$, $a$, and $b$ here is a string and may contain multiple tokens. Generation is performed with temperature 1.0 and nucleus sampling with $p = 0.9$ (Holtzman et al., 2019).

## 4    Attention Decay: a Hypothesis

It is reasonable to hypothesize that instruction drift results from a decaying influence of the prompt over time. To investigate why this happens, we focus on the attention distribution over context tokens in transformer self-attention heads. Although the intuitive hypothesis broadly captures the underlying phenomenon, our empirical and theoretical analyses uncover nuanced discrepancies.

### 4.1    Preliminaries

Suppose the input tokens are $\{w_i\}_{i=1}^t$, each belonging to the vocabulary $V$. To generate the next token $w_{t+1} \in V$, the current tokens are first embedded into $D$-dimensional vectors $\{h_i^0\}_{i=1}^t$ with the embedding matrix $W_e \in \mathbb{R}^{|V| \times D}$. These are then processed sequentially by $L$ transformer layers, resulting in a grid of activations after each layer and for each token

---

[†]Omitting formatting tokens like , <<SYS>> or [INST].

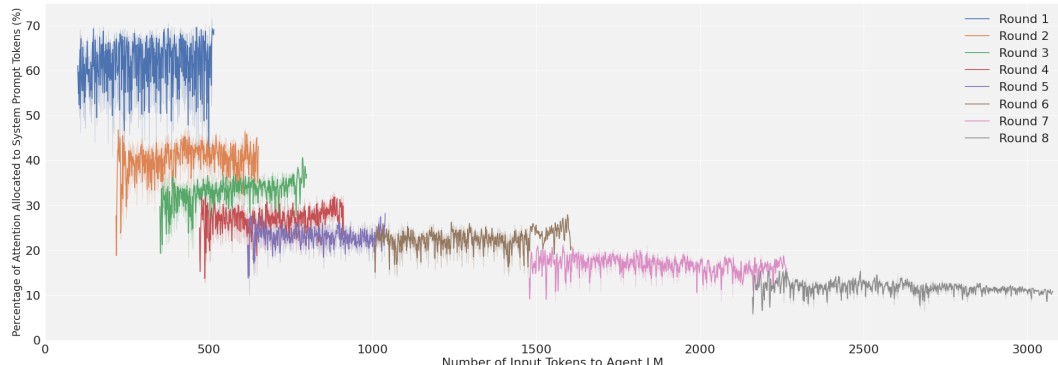

Figure 4: The phenomenon of attention decay demonstrated in the 11th attention head in the 24th layer of LLaMA2-7B, which has a maximum context window size of 4,096 tokens. We generate 12 conversations while tracking the portion of attention allocated to system prompt tokens. The plots are specifically for the agent LM, grouped by the rounds in which the answers are generated; the values are absent for the user LM. We observe sharp drops in attention between turns and rough plateaus within turns.

$\{h_i^l\}_{i=1,l=1}^{t,L}$. As the multi-layer perception (MLP) and layer norm are context-independent, we leave them out for simplicity. The feed-forward process of the transformer can be summarized as:

$$h_i^l = h_i^{l-1} + \sum_{m=1}^{H} W_o^{l,m} \text{Att}^{l,m}(h_1^{l-1}, \dots, h_i^{l-1}), \tag{1}$$

$$w_{t+1} \sim p(w|w_{\leq t}) = \text{softmax}(W_e h_t^L). \tag{2}$$

The combination of the softmax and $W_e$ work as a predictor from $h_t^L$ to distribution $p(w|w_{\leq t})$ of next token $w_{t+1}$. $\text{Att}^{l,m}$ is the single head attention operator with output in a lower dimensional space and $W_o^{l,m} \in \mathbb{R}^{D \times d}$ maps them back into $\mathbb{R}^D$, the residual stream space.

Crucial to our experiment, we expand the attention operator to show it aggregates activations from previous time steps based on an attention distribution:

$$\alpha_{t,j=1:t}^{l,m} = \text{softmax}\left(\frac{(W_k^{l,m} h_{1:t}^{l-1})^\top (W_q^{l,m} h_t^{l-1})}{\sqrt{d}}\right). \tag{3}$$

Then the attention operation is a weighted sum of linearly transformed activations from the last layer:

$$\text{Att}^{l,m}(h_1^{l-1}, \dots, h_t^{l-1}) = \sum_{j=1}^{t} \alpha_{t,j}^{l,m} \left(W_v^{l,m} h_j^{l-1}\right), \tag{4}$$

where $W_v^{l,m} \in \mathbb{R}^{d \times D}, W_k^{l,m} \in \mathbb{R}^{d \times D}, W_q^{l,m} \in \mathbb{R}^{d \times D}$ are the value, key, and query weight matrices, respectively.

## 4.2 The Phenomenon of Attention Decay

While generating the next token given an input sequence containing $t$ tokens, in each attention head, the last token will compute a normalized attention distribution over all previous tokens (including itself), denoted by $\alpha_{t,i=1:t}$ in Equation 3. Tokens in the system prompt are a special subset of all previous tokens, and we denote the sum of the attention weights allocated to them as $\pi(t) = \sum_{i=1}^{|s_B|} \alpha_{t,i}$. It ranges between 0 to 1 and represents the comparative importance that the system prompt has throughout the generation process. We monitor this percentage $\pi(t)$ along the decoding time steps $t$ and across turns of conversations in LLaMA2-7B. We only plot $\pi(t)$ from the perspective of the agent LM.

As shown in Figure 4, within each turn, $\pi(t)$ remains almost constant, but there are significant decreases across turns. This observation runs over a naive hypothesis of attention decay—if the attention distributes uniformly over previous tokens, $\pi(t)$ should decay hyperbolically and be independent of number of turns.

It's also worth-noting that this highlights a unique issue in chatbots, distinct from language models, where out-of-distribution text from interlocutors is absent. The case of the language model completing its input partial sequence is technically equivalent to the agent LM generating answers for a single turn, which displays a plateau in $\pi(t)$.

This observation shows merely the co-occurrence of instruction drift and attention decay. However, it inspires the hypothesis that attention decay may internally contribute to instruction drift, suggesting that addressing the former could help mitigate the latter (Section 6.2).

## 5   A Geometric View of Attention Decay

To shed light on attention decay in Figure 4, both the plateau within utterance and the drop across utterances, we provide a theoretical explanation in a simplified situation. It has been shown empirically and theoretically that the internal representation of deep neural networks usually live in a narrow cone in the high-dimensional space (Mimno & Thompson, 2017; Ethayarajh, 2019; Zhu et al., 2021; Liang et al., 2022). Motivated by their observations, we characterize attention decay from a similar geometric perspective.

We will consider two settings of model generation:

1. New tokens are generated autoregressively given initial tokens $h_1, \ldots, h_{|s_B|}$, which models the process of the agent LM generating answers;

2. New tokens are drawn by the user. A user LM could put out-of-distribution tokens into the context window of agent LM in a potentially adversarial fashion (Zou et al., 2023).

For the first setting, we will show that tokens generated by the model always remain in an approximately low-dimensional convex cone in Theorem 5.1. In the second setting, we can characterize the expansion using spherical measure and show that randomly drawn tokens will lead to an expansion of the underlying convex cone with the growth of intrinsic dimension of token embeddings, as shown in Proposition A.2 in Appendix A.

### 5.1   Setting One: Agent Utterances

In linear algebra, a *cone* is a subset of a vector space that is closed under positive scalar multiplication. In other words, $C$ is a cone if $x \in C$ implies $sx \in C$ for every positive scalar $s$. Moreover, $C$ is called a *convex cone* if $\alpha x + \beta y \in C$ for any positive scalars $\alpha$ and $\beta$, and any $x, y \in C$.

The dimension of a cone is the dimension of the vector space spanned by the elements of the cone. For convenience, we define two new notions related to low dimensional cones in the space $\mathbb{R}^D$. Given any $d$-dimensional convex cone $C \subset \mathbb{R}^D$ ($1 \leq d \leq D$), for $\epsilon \in (0, 1)$ we define the corresponding $\epsilon$-*approximate $d$-dimensional cone* as

$$C^\epsilon := \{w \in C \oplus \mathrm{span}(C)^\perp \subset \mathbb{R}^D : w = u + v$$
$$\text{for some } u \in C, v \in \mathrm{span}(C)^\perp \cong \mathbb{R}^{D-d}, \|v\| \leq \epsilon\|w\|\}.$$

Given some $c \in \mathbb{S}^{D-1}$ and $\theta \in (0, \pi/2)$, a $d$-dimensional *spherical cone* is the set defined by

$$P^d[c, \theta] := \{u \in U \subset \mathbb{R}^D : U \cong \mathbb{R}^d, \langle c, u \rangle \geq \|u\| \cos \theta\}.$$

**Theorem 5.1.** *Assume that the token embeddings of the system prompt given by $h_1, \ldots, h_{|s_B|}$ lie in the $d$-dimensional approximate cone $C^\epsilon$, and that any output-value matrix $W_{ov}^{l,m} = W_o^{l,m} W_v^{l,m} \in \mathbb{R}^{D \times D}$ satisfy that $W_{ov}^{l,m} u \in C^\epsilon$ for any $u \in C^\epsilon$. Then all proceeding tokens generated by our*

*simplified transformer lie in the convex hull of $C^\epsilon$. In particular, if $C^\epsilon$ is contained in some spherical cone $P^d[c, \theta]$, then all generated tokens lie in the $\tilde{\epsilon}$-approximate cone $C^{\tilde{\epsilon}}$ where $\tilde{\epsilon} = \epsilon / \sqrt{\epsilon^2 + \cos^2 \theta (1 - \epsilon^2)}$.*

For the initial tokens, $\theta$ indicates how concentrated their embeddings are, and $d$ is roughly the intrinsic dimension of these embeddings. Note that $d \leq |s_B|$ and the number of tokens in the system prompt $|s_B|$ is usually much smaller than the dimensions of hidden space $D$, which is 8192 in the case of LLaMA2-70B-chat. Thus, the assumption that initial embeddings occupy a low-dimensional cone is reasonable.

Theorem 5.1 shows the convex cone for token embeddings remains stable during the generating process if there is no user input, which leads to the plateau within an utterance.

# 6 Mitigating Instruction Drift

If instruction drift is related to attention decay, that suggests we can mitigate drift by manipulating the level of attention on the original prompt. Before presenting an attention-based mitigation method, however, we describe two baselines.

## 6.1 Baseline Methods

**System Prompt Repetition (SPR)**  We inject the system prompt with probability $0 \leq p \leq 1$ before each user utterance. The repeated system prompts, like the standard system prompt at the start of the input sequence, only appear when the language model is prompted; users do not see them.

**Classifier-Free Guidance (CFG)**  The second method is classifier-free guidance (CFG, Sanchez et al., 2023), which runs the base model twice, firstly with system prompt to get $\log p(w|w_{\leq t}, s_B)$ and then without system prompt to get $\log p(w|w_{\leq t})$. It then uses a contrastive linear operation inside the logit space to strengthen the effects of the system prompt on answer generation. The new next-token probability distribution is defined by:

$$\log \hat{p}(w|w_{\leq t}, s_B) = \log p(w|w_{\leq t}) + \alpha(\log p(w|w_{\leq t}, s_B) - \log p(w|w_{\leq t})). \tag{5}$$

CFG comes with a hyperparameter $\alpha \geq 1$ that controls how far we shift the predicted logits. When $\alpha = 1$, it reduces to prompting with the system prompt; larger $\alpha$ produces stronger intervention.

## 6.2 Proposed Method: Split-softmax (SS)

Motivated by the attention decay phenomenon, we introduce a method that requires no retraining, **split-softmax**, aimed at reducing this decay with minimal overhead. The basic idea is straightforward: if the problem is that the model pays too little attention to the prompt, then force the model to pay more. In practice, we find that a power-law scaling of attention seems to be effective.

In particular, split-softmax (SS) works by inserting a scaling operation between Equation 3 and Equation 4 for every attention operation. After obtaining the attention distribution $\{\alpha_{t,i}\}_{i=1}^t$ which sums up to 1 (omitting superscript for simplicity), we reweight it by:

$$\pi(t) = \sum_{i=1}^{|s_B|} \alpha_{t,i}, \quad \alpha'_{t,i} = \begin{cases} \frac{\pi^k(t)}{\pi(t)} \alpha_{t,i} & \text{if } i \leq |s_B| \\ \frac{1 - \pi^k(t)}{1 - \pi(t)} \alpha_{t,i} & \text{if } i > |s_B| \end{cases}, \tag{6}$$

where the introduced exponent $0 \leq k \leq 1$ as a hyperparameter to control the strength of our intervention. The smaller $k$ is, the stronger the intervention is; when $k = 1$, the intervention is nullified. The new set of attention $\{\alpha'_{t,i}\}_{i=1}^t$ sums up to 1 as well and will replace $\{\alpha_{t,i}\}_{i=1}^t$ so that more attention is paid to the system prompt tokens. Given $0 \leq \pi(t) \leq 1, 0 \leq k \leq 1$ thus $\frac{\pi^k(t)}{\pi(t)} \geq 1$, split-softmax increases the proportion of attention paid to system prompts. See Appendix E for more discussion.

## 6.3 Calibration Using Performance Drop on MMLU

Each method (split-softmax and the two baselines) represents a potentially large intervention; any instruction stabilization may come at the expense of other capabilities of the model. However, each method has a hyperparameter that corresponds to the strength of the intervention. To compare methods, therefore, we need to measure both the increase in instruction stability and the performance drop for various values of the relevant hyperparameter. This is analogous to measuring a precision-recall curve for a classifier.

To measure any performance changes, we use the Massive Multitask Language Understanding (MMLU, Hendrycks et al., 2020). To compare the different methods, look at the stability improvement at equal levels of performance drop. Sweeping hyperparameters for each method allows us to measure and plot each method's stability-performance curve, revealing different trade-offs between our stability metric and MMLU performance.

As expected, we do see an inverse relationship between performance and instruction stability in all three of our methods Figure 5. This corroborates earlier findings by Gu et al. (2024) that control methods over language model often come at the cost of general capability. The performance drop on MMLU should be thought of as a budget when correcting model behaviors, and two methods should only be compared on stability when their respective hyperparameters cause similar MMLU performance drop.

To quantify stability, we use a 16-turn conversation as described in Figure 2. We modify these conversations by applying each method to the agent LM.

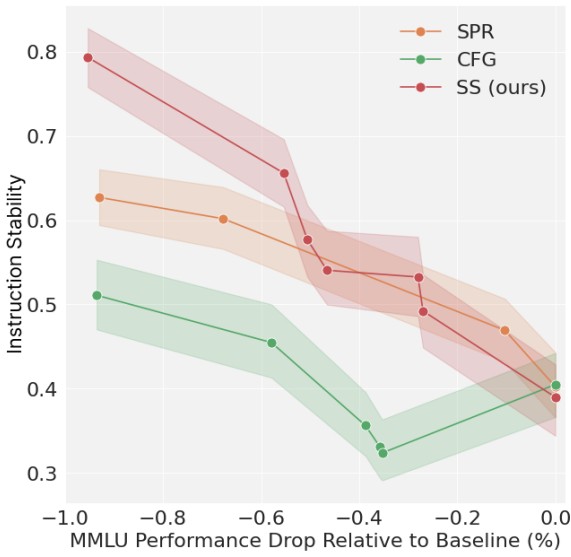

Figure 5: **Comparing trade-offs between instruction stability and performance.** For each of the three methods, we vary a hyperparameter that reflects the strength of the intervention. Each curve plots the effect on stability and performance over the hyperparameter sweep. Compared to two baselines (classifier-free guidance and system prompt repetition), split-softmax produces equal or higher stability for a given level of performance degradation.

Then we probe the agent LM at each round to test its instruction stability in the same fashion as section 3. Stability is measured for individual turns, and the overall stability measure is the average of the stability at each turn of agent LM. Given the conversation history of agent LM under intervention, we sample one and ask questions from MMLU at an intermediate turn (the 4th turn in our experiments); and the answers are used to calculate MMLU accuracy. Note that due to the added system prompt and chat history, the MMLU performance is different from what is reported by LLaMA team even without intervention (Touvron et al., 2023). However, only the difference between post- and pre-intervention performances is meaningful, as the primary purpose of using MMLU in our case is to calibrate the strength of the intervention.

## 6.4 Experimental Results

All experiments are conducted on LLaMA2-70B-chat. To save computational cost, we choose one system prompt from each of the five categories, and run experiments over the total twenty ordered pairs of system prompts.

In Figure 5 we plot instruction stability versus performance drop on MMLU as we vary the strength hyperparameter for each method. In general, split-softmax presents a better

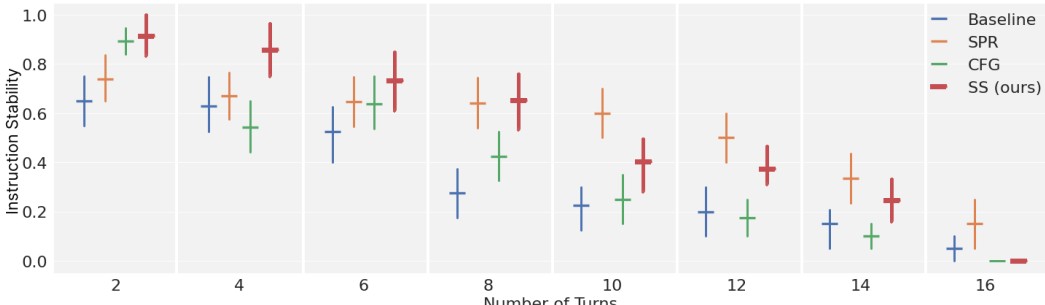

Figure 6: Comparison of instruction stability across turns, with MMLU performance drop around the value of 0.5, for system prompt repetition (SPR), classifier-free guidance (CFG), and split-softmax (SS). The whisker represents one standard deviation.

trade-off between performance drop and instruction stability. It can match performance with system prompt repetition while avoiding using the additional context window. If more drop in performance on MMLU is allowed, split-softmax enables greater instruction stability.

In Figure 6, we break down the instruction stability measurement across turns. Similar to what Sanchez et al. (2023) show, classifier-free guidance helps the model adhere to the system prompt remarkably well for the first round of the conversation, but it does not generalize well into extended conversations. Both system prompt repetition and split-softmax demonstrate higher effectiveness in mitigating instruction drift, though they exhibit different trends. The former excels in regions with a larger number of turns, while the latter performs better at the beginning of the conversation. Note that system prompt repetition consumes a substantial portion of the context window.

## 7 Conclusions, Discussions, and Future Work

Our experiments indicate that instruction drift is a potentially significant issue for prompt engineering. To help address this challenge, we contribute a new protocol and benchmark to help measure this phenomenon, as well as an idealized mathematical model of its cause. In addition, we proposed a technique, split-softmax, that can help mitigate instruction drift, providing a better stability-performance trade-off than two existing baselines.

The instruction drift indicates that current LM systems still cannot behave coherently over long horizons, which is the basis of long-term planning, such as information seeking (Lin et al., 2023) and tool usage (Nakano et al., 2021). Fundamentally, it is caused by a mismatch between their training schemes (text continuation or single-round RLHF) and their deployment scenarios (open-ended dialog with users). The phenomenon of instruction drift suggests there is much to be done to bridge this gap toward more robust and coherent dialog systems.

There is ample room for future work in this space. For example, it would be natural to explore making changes in architecture or to training to combat instruction drift. Furthermore, all the techniques we discussed involve an apparent trade-off between performance and reliability. Is this a necessary compromise, or are there methods that maintain instruction stability at no cost? It would also be good to deepen our theoretical understanding, adding realism to the idealized "cone" model of instruction drift that we proposed. Finding new ways to measure and prevent instruction drift is an important step in ensuring AI safety and reliability.

## Acknowledgments

We thank Jiawei Zhou for useful discussions and feedback on the manuscript.

KL is supported by a fellowship from the Kempner Institute for the Study of Natural and Artificial Intelligence at Harvard University and Superalignment Fast Grants from OpenAI. DB is supported by a grant from Open Philanthropy. This work has been made possible in part by a gift from the Chan Zuckerberg Initiative Foundation to establish the Kempner Institute for the Study of Natural and Artificial Intelligence. This work was partially supported by NSF grant IIS-1901030.

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

# APPENDIX

## A  Sketch of the Theory for Setting Two: User Utterances ( Section 5)

Again we assume that the system tokens $h_1, \ldots, h_{\|s_B\|}$ are from some $C_0^\epsilon$, and let $C_n$ be the smallest convex cone containing $C_0$ and user tokens $\{h_{|s_B|+i}\}_{i=1}^n$. Then the expansion $C_0 \subset C_1 \subset \cdots \subset C_n$ reflects the attention decay under the influence of user utterances. To get some intuition on the expanding process, we show the following:

**Proposition A.1.** *If user tokens are drawn i.i.d. uniformly from $\mathbb{S}^{D-1}$, then with probability $1 - \eta$ after $n \geq 4D + 2\log\frac{1}{\eta}$ user tokens $C_n$ expands to the whole space $\mathbb{R}^D$.*

Proposition A.1 suggests that when user utterances are inserted, the size of the convex cone for token embeddings will grow significantly, which gives rise to the drop of $\pi(t)$ across utterances. To further quantify the expansion of convex cones, we can consider the spherical measure $\sigma_{D-1}$, which is the Borel measure on the $(D-1)$-sphere such that $\sigma_{D-1}(\mathbb{S}^{D-1}) = 1$. For any $\epsilon$-approximate convex cone $C^\epsilon$, define the volume of $C^\epsilon$ by

$$\mu(C^\epsilon) := \sigma_{D-1}(C^\epsilon \cap \mathbb{S}^{D-1}).$$

Then intuitively $\mu(C_0^\epsilon)/\mu(C_n^\epsilon)$ indicates the degree to which the current tokens in $C_n^\epsilon$ align with the system tokens in $C_0^\epsilon$, similar to the quantity $\pi(t)$ defined in the previous section.

In real applications, user messages are not i.i.d. uniform variables from $\mathbb{S}^{D-1}$. However, there usually exists an evident proportion of user tokens distinct from the system tokens. They could probably be tokens unique in the specific topics that the user inquires about or, more typically, tokens from a new language. It could also happen that the user is attacking the LM by sending adversarial tokens (Zou et al., 2023). The following proposition quantifies how attention decays in terms of $\mu(C_0^\epsilon)/\mu(C_n^\epsilon)$ as such embedding dimension increases.

**Proposition A.2.** *Suppose $C_0$ is a $d_1$-dimensional convex cone contained in some $d_1$-dimensional spherical cone $P^{d_1}[c_1, \psi_1]$ while $C_n$ is a $d_2$-dimensional convex cone containing a $d_2$-dimensional spherical cone $P^{d_2}[c_2, \psi_2]$. Then we have*

$$\frac{\mu(C_0^\epsilon)}{\mu(C_n^\epsilon)} \lesssim \epsilon^{d_2 - d_1}.$$

The geometric perspective we proposed provides a concrete explanation of why inserting user prompts will cause attention decay while autoregressive generation from the model will almost have no harm. However, one limitation here is that we have only compared the cone structure without tracking the distribution of token embeddings within the cones. In particular, if we force the majority of tokens generated from $C_n^\epsilon$ to be contained or close to $C_0^\epsilon$, the issue of attention decay could possibly be mitigated, which motivates our method in the proceeding section.

## B  Proofs for Theorems

We start by making simplifications to the model and token-generating process. First, the model is simplified by omitting the MLP and layer norms as in Equation 1. For the token-generating process, the embedding of the next token $h_{t+1}$ is close to $h_t^L$ among all tokens in the vocabulary in Equation 2. Thus, for convenience we directly put $h_{t+1} := h_t^L/\|h_t^L\|$ in our simplified model, meaning that all embeddings lie on the unit hypersphere $\mathbb{S}^{D-1} := \{v \in \mathbb{R}^D : \|v\| = 1\}$.

*Proof of Theorem 5.1.* Let $\overline{C^\epsilon}$ be the convex hull of $C^\epsilon$. The $\overline{C^\epsilon}$ is a convex cone containing $C^\epsilon$. Theorem 5.1 can be proven in two steps.

Step I. We establish that $h_t \in \overline{C^\epsilon}$ by induction. $h_1, \ldots, h_{t_0}$ already satisfy the claim by assumption. Supposing that $h_1, \ldots, h_t \in \overline{C^\epsilon}$ $(t \geq t_0)$, we show that $h_{t+1}$ is also in $\overline{C^\epsilon}$. Here we look into $h_j^l$ $(j = 1, \ldots, t, l = 1, \ldots, L)$ in the process of generating $h_{t+1}$. We perform induction on $l$. For $l = 0$, we have $h_j^l = h_j \in \overline{C^\epsilon}$. Supposing that $h_j^l \in \overline{C^\epsilon}$ for $j = 1, \ldots, t$, it suffices to prove that $h_j^{l+1} \in \overline{C^\epsilon}$.

By induction hypothesis that $h_j^l \in \overline{C^\epsilon}$ $(j = 1, \ldots, t)$ we can find $k_j \in \mathbb{N}^+$, $x_{j,1}, \ldots, x_{j,k_j} \in C^\epsilon$, and $w_{j,1}, \ldots, w_{j,k_j} > 0$ for $j = 1, \ldots, t$ such that

$$h_j^l = \sum_{i=1}^{k_j} w_{j,i} x_{j,i}.$$

Thus, by Equation 1 we have

$$
\begin{aligned}
h_j^{l+1} &= h_j^l + \sum_{m=1}^{H} W_o^{l+1,m} \mathrm{Att}^{l+1,m}(h_1^l, \ldots, h_j^l) \\
&= h_j^l + \sum_{m=1}^{H} \sum_{s=1}^{j} \alpha_{j,s}^{l+1,m} W_o^{l+1,m} W_v^{l+1,m} h_s^l \\
&= h_j^l + \sum_{m=1}^{H} \sum_{s=1}^{j} \sum_{i=1}^{k_s} \alpha_{j,s}^{l+1,m} w_{s,i} W_o^{l+1,m} W_v^{l+1,m} x_{s,i}.
\end{aligned}
$$

Note that $\alpha_{j,s}^{l+1,m} > 0$ since it is calculated from softmax and by assumption we have $W_o^{l+1,m} W_v^{l+1,m} x_{i,s} \in C^\epsilon$ as $x_{s,i} \in C^\epsilon$. Thus, we conclude that $h_j^{l+1} \in \overline{C^\epsilon}$. By induction we know for $l = 1, \ldots, L$ and $j = 1, \ldots, t$ we have $h_j^l \in \overline{C^\epsilon}$. Thus, $h_{t+1} = h_t^L / \|h_t^L\| \in \overline{C^\epsilon}$ holds. And by induction again we conclude that $h_t \in \overline{C^\epsilon}$ for all $t \geq 1$.

Step II. Let $\gamma = \cos\theta$. We prove that $\overline{C^\epsilon} \subset C^{\tilde{\epsilon}}$ where $\tilde{\epsilon} = \epsilon / \sqrt{\epsilon^2 + \gamma^2(1 - \epsilon^2)}$. For any $y \in \overline{C^\epsilon}$, there exists $k \in \mathbb{N}^+$, $x_1, \ldots, x_k \in C^\epsilon$, and $w_1, \ldots, w_k > 0$ such that $y = \sum_{i=1}^{k} w_i x_i$. By definition of $C^\epsilon$, $x_i$ can be written as $x_i = u_i + v_i$ where $u_i \in C$ and $v_i \in \mathrm{span}(C)^\perp$ and $\|v_i\| \leq \epsilon \|x_i\|$. By definition of $P^d[c, \theta]$ we have $\langle c, u_i \rangle \geq \gamma \|u_i\|$ for all $i = 1, \ldots, k$. Let $\tilde{u}_i := \langle c, u_i \rangle c$. Then $\langle \tilde{u}_i, u_i - \tilde{u}_i \rangle = 0$ and hence $\langle \sum_{i=1}^{k} w_i \tilde{u}_i, \sum_{i=1}^{k} w_i(u_i - \tilde{u}_i) \rangle = 0$. Therefore, we have

$$\left\| \sum_{i=1}^{k} w_i u_i \right\| \geq \left\| \sum_{i=1}^{k} w_i \tilde{u}_i \right\| = \sum_{i=1}^{k} \left\langle c, \sum_{i=1}^{k} w_i u_i \right\rangle \geq \gamma \sum_{i=1}^{k} w_i \|u_i\|.$$

On the other hand, we know

$$\left\| \sum_{i=1}^{k} w_i v_i \right\| \leq \sum_{i=1}^{k} w_i \|v_i\| \leq \frac{\epsilon}{\sqrt{1 - \epsilon^2}} \sum_{i=1}^{k} w_i \|u_i\|.$$

Therefore, it holds that

$$\left\| \sum_{i=1}^{k} w_i u_i \right\| \geq \frac{\gamma \sqrt{1 - \epsilon^2}}{\epsilon} \left\| \sum_{i=1}^{k} w_i v_i \right\|,$$

which implies that

$$\left\| \sum_{i=1}^{k} w_i v_i \right\| \geq \frac{\epsilon}{\sqrt{\epsilon^2 + \gamma^2(1 - \epsilon^2)}} \left\| \sum_{i=1}^{k} w_i x_i \right\|.$$

Thus, we conclude that $\overline{C^\epsilon} \subset C^{\tilde{\epsilon}}$. $\qquad \square$

To prove Proposition A.1 we need the following lemma.

**Lemma B.1** (Wendel, 1962). *Let N points be scattered uniformly at random on $\mathbb{S}^m \subset \mathbb{R}^{m+1}$. Then the probability that all points lie on some hemisphere is given by*

$$a_{m,N} = 2^{-N+1} \sum_{k=0}^{m} \binom{N-1}{k}.$$

*Proof of Proposition A.1.* If there is no hemisphere containing $h_{t_0+1}, \ldots, h_{t_0+n}$, then the origin lies in $C_n$ and is not on the boundary, meaning that $C_n = \mathbb{R}^D$. Thus, we only need to show that for $n \geq 4D + \log \frac{1}{\eta}$, it holds that $a_{D,n} \leq \eta$. Since

$$2^{-n} \sum_{i=0}^{D} \binom{n}{i} \leq 2^{-n} \sum_{i=0}^{D} \frac{n^i}{i!} = 2^{-n} \sum_{i=0}^{D} \frac{D!}{i!} \left(\frac{n}{D}\right)^i \leq 2^{-n} \left(\frac{en}{D}\right)^D.$$

It suffices to prove that $2^{-n} \left(\frac{en}{D}\right)^D < \eta$. For convenience let $\alpha := 4 + \frac{2}{D} \log \frac{1}{\eta} \leq \frac{n}{D}$. Then we can check that

$$\left(\log 2 - \frac{1}{2}\right) e^{\alpha/2} > \left(\frac{1}{\eta}\right)^{1/D}.$$

Note that

$$e^{\alpha(\log 2 - \frac{1}{2}) - 1} \geq \alpha \left(\log 2 - \frac{1}{2}\right),$$

which is equivalent to

$$e\alpha \leq \frac{e^{\alpha(\log 2 - \frac{1}{2})}}{\log 2 - \frac{1}{2}} = \frac{2^\alpha}{e^{\alpha/2} \left(\log 2 - \frac{1}{2}\right)}.$$

Thus, we have

$$2^{-n} \left(\frac{en}{D}\right)^D \leq \frac{(e\alpha)^D}{2^{\alpha D}} \leq \frac{1}{\left(\log 2 - \frac{1}{2}\right)^D e^{\alpha D/2}} < \eta.$$

$\square$

To show Proposition A.2 we need the following lemma.

**Lemma B.2** (Li, 2010). *The spherical measure of the spherical cap $P^{m+1}[c, \theta] \cap \mathbb{S}^m$ is given by*

$$\sigma_m(P^{m+1}[c, \theta] \cap \mathbb{S}^m) = \frac{\int_0^\theta \sin^{m-1} x \, dx}{2 \int_0^{\pi/2} \sin^{m-1} x \, dx} = \frac{\Gamma(\frac{m+1}{2})}{\sqrt{\pi} \Gamma(\frac{m}{2})} \int_0^\theta \sin^{m-1} x \, dx,$$

*where $\Gamma(x)$ is the Gamma function.*

*Proof of Proposition A.2.* First we lower bound $\mu(C_n^\epsilon)$ by identifying as many disjoint spherical caps with angle $\theta := \arcsin \epsilon$ as possible and applying Lemma B.2.

Let $M$ be the largest number such that there exists a set of points $a_1, \ldots, a_M \in P^{d_2}[c_2, \psi_2 - \theta] \cap \mathbb{S}^{D-1}$ to ensure $P^D[a_i, \theta] \subset P^{d_2}[c_2, \psi_2]$ ($i = 1, \ldots, M$) are disjoint from one another ("disjoint" meaning that the measure of intersection is zero). We claim that $\{P^{d_2}[a_i, 2\theta]\}_{i=1}^{M}$ is a covering of $P^{d_2}[c_2, \psi_2]$. Otherwise, choosing $a_0 \in P^{d_2}[c_2, \psi_2] \cap \mathbb{S}^{D-1} \setminus \bigcup_i P^{d_2}[a_i, 2\theta]$ we can check that $P^D[a_0, \theta]$ does not intersect with any of $P^D[a_i, \theta]$. Thus, these $M + 1$ spherical caps do not overlap, which contradicts the definition of $M$. Hence $P^{d_2}[c_2, \psi_2] \subset \bigcup_i P^{d_2}[a_i, 2\theta]$, and by Lemma B.2 we have

$$\frac{\Gamma(\frac{d_2}{2})}{\sqrt{\pi} \Gamma(\frac{d_2-1}{2})} \int_0^{\psi_2} \sin^{d_2-2} x \, dx = \sigma_{d_2-1}(P^{d_2}[c_2, \psi_2] \cap \mathbb{S}^{D-1})$$

$$\leq \sum_{i=1}^{M} \sigma_{d_2-1}(P^{d_2}[a_i, 2\theta] \cap \mathbb{S}^{D-1}) = M \sigma_{d_2-1}(P^{d_2}[a_i, 2\theta]) = M \frac{\Gamma(\frac{d_2}{2})}{\sqrt{\pi} \Gamma(\frac{d_2-1}{2})} \int_0^{2\theta} \sin^{d_2-2} x \, dx.$$

On the other hand, since $P^D[a_i, \theta]$'s are disjoint from each other and that $P^D[a_i, \theta] \subset P^D[c_2, \psi_2]$ (because $\epsilon = \sin \theta$), we know

$$
\begin{aligned}
\mu(C_n^\epsilon) &\geq \sum_{i=1}^M \sigma_{D-1}(P^D[a_i, \theta] \cap \mathbb{S}^{D-1}) = M\sigma_{D-1}(P^D[a_i, \theta] \cap \mathbb{S}^{D-1}) \\
&= M\frac{\Gamma(\frac{D}{2})}{\sqrt{\pi}\Gamma(\frac{D-1}{2})} \int_0^\theta \sin^{D-2} x dx \\
&\geq \frac{\Gamma(\frac{D}{2})}{\Gamma(\frac{D-1}{2})} \frac{\int_0^{\psi_2} \sin^{d_2-2} x dx \int_0^\theta \sin^{D-2} x dx}{\int_0^{2\theta} \sin^{d_2-2} x dx}.
\end{aligned}
$$

Next we upper bound $\mu(C_0^\epsilon)$. For any $(x_1, \cdots, x_n) \in \mathbb{B}^n := \{(x_1, \ldots, x_n) : \sum_{i=1}^n x_i^2 \leq 1\}$, we introduce the hyperspherical coordinate system, which consists of a radial coordinate $r$, and $n-1$ angular coordinates $\phi_1, \ldots, \phi_{n-1}$, where the angles $\phi_1, \cdots, \phi_{n-2}$ range over $[0, \pi]$ and $\phi_{n-1}$ ranges over $[0, 2\pi)$. In specific, the coordinates are defined through the transformation:

$$
\begin{aligned}
x_1 &= r \cos \phi_1, \\
x_2 &= r \sin \phi_1 \cos \phi_2, \\
x_3 &= r \sin \phi_1 \sin \phi_2 \cos \phi_3, \\
&\vdots \\
x_{n-1} &= r \sin \phi_1 \cdots \sin \phi_{n-2} \cos \phi_{n-1}, \\
x_n &= r \sin \phi_1 \cdots \sin \phi_{n-2} \sin \phi_{n-1}.
\end{aligned}
$$

By assumption we know $C_0 \subset P^D[c_1, \psi_1]$. Therefore, using the notion of spherical elements (Blumenson, 1960), we can write

$$
\mu(C_0^\epsilon) = \sigma_{D-1}(C_0^\epsilon \cap \mathbb{S}^{D-1}) = \frac{1}{\text{Area}(\mathbb{S}^{D-1})} \int_\Omega \sin^{D-2} \phi_1 \sin^{D-3} \phi_2 \cdots \sin \phi_{D-2} d(\phi_1, \ldots, \phi_{D-1}),
$$

where

$$
\Omega = \left\{ (\phi_1, \cdots, \phi_{D-1}) : \phi_1 \in [0, \psi_1], \phi_2, \ldots, \phi_{D-2} \in [0, \pi], \phi_{D-1} \in [0, 2\pi], \prod_{j=1}^{d_1-1} \sin \phi_j \in [0, \epsilon] \right\}.
$$

Denoting

$$
\Omega_1 = \left\{ (\phi_1, \cdots, \phi_{d_1-1}) : \phi_1 \in [0, \psi_1], \phi_2, \ldots, \phi_{d_1-1} \in [0, \pi], \prod_{j=1}^{d_1-1} \sin \phi_j \in [0, \epsilon] \right\},
$$

then we have

$$
\begin{aligned}
\mu(C_0^\epsilon) &= \frac{1}{\text{Area}(\mathbb{S}^{D-1})} \int_{(\phi_1,\ldots,\phi_{d_1-1}) \in \Omega_1} \sin^{D-2} \phi_1 \cdots \sin^{D-d_1} \phi_{d_1-1} d(\phi_1, \ldots, \phi_{d_1-1}) \\
&\qquad \int_0^\pi \cdots \int_0^\pi \int_0^{2\pi} \sin^{D-d_1-1} \phi_{d_1} \cdots \sin \phi_{D-2} d\phi_{d_1} \cdots d\phi_{D-1} \\
&= \frac{\text{Area}(\mathbb{S}^{D-d_1})}{\text{Area}(\mathbb{S}^{D-1})} \int_{(\phi_1,\ldots,\phi_{d_1-1}) \in \Omega_1} \sin^{D-2} \phi_1 \cdots \sin^{D-d_1} \phi_{d_1-1} d(\phi_1, \ldots, \phi_{d_1-1}) \\
&\leq \frac{\text{Area}(\mathbb{S}^{D-d_1})}{\text{Area}(\mathbb{S}^{D-1})} \epsilon^{D-d_1} \int_0^{\psi_1} \int_0^\pi \cdots \int_0^\pi \sin^{d_1-2} \phi_1 \cdots \sin \phi_{d_1-2} d\phi_1 \cdots d\phi_{d_1-1} \\
&= \frac{\text{Area}(\mathbb{S}^{D-d_1})\text{Area}(\mathbb{S}^{d_1-1})}{2\text{Area}(\mathbb{S}^{D-1})} \sigma_{d_1-1}(P^{d_1}[c_1, \psi_1] \cap \mathbb{S}^{D-1}) \\
&= \frac{\Gamma(\frac{D}{2})}{\Gamma(\frac{D-d_1+1}{2})\Gamma(\frac{d_1-1}{2})} \epsilon^{D-d_1} \int_0^{\psi_1} \sin^{d_1-2} x dx.
\end{aligned}
$$

Thus, we conclude that

$$\frac{\mu(C_0^\epsilon)}{\mu(C_n^\epsilon)} \leq \frac{\Gamma(\frac{D-d_1+1}{2})\Gamma(\frac{d_1-1}{2})}{\Gamma(\frac{D-1}{2})} \frac{\int_0^{\psi_1} \sin^{d_1-2} x dx \int_0^{2\arcsin\epsilon} \sin^{d_2-2} x dx}{\int_0^{\psi_2} \sin^{d_2-2} x dx \int_0^{\arcsin\epsilon} \sin^{D-2} x dx} \epsilon^{D-d_1}$$

$$\lesssim \epsilon^{D-d_1} \frac{\epsilon^{d_2-1}}{\epsilon^{D-1}} = \epsilon^{d_2-d_1}.$$

$\square$

Figure 7: Histogram of embedding vector norms.

**Norm of Embedding Vectors**  In Section 3.2, we assume that the embedding vectors have the unit norm. To verify if this is reasonable, we plot the density of the norms of vocabulary embeddings for the LLaMA2-7B-chat in Figure 7. We can observe that the norms are quite concentrated around 1.

## C  Does RLHF help?

Given how RLHF Ouyang et al. (2022); Ziegler et al. (2019) train the model, the model should be trained to pay more attention to the system prompt so to increase user satisfaction. In Figure 8, we show that RLHF could increase the portion of attention paid to the system prompts by comparing LLaMA2-7B and LLaMA2-7B-chat. The latter is trained on top of the former with human feedback. It shows that RLHF indeed helps in combating instruction drift, but it still cannot eradicate it entirely due to its nature of fine-tuning.

## D  Additional Instruction Drift Experiments

To see how close-source model compares with LLaMA2-70B-chat, we test gpt-3.5-turbo-16k with a total of 200 randomly sampled system prompt pairs. Results are shown in Figure 9. It turns out that gpt-3.5-turbo-16k holds to its system prompt better than LLaMA2-chat-70B, but still suffers a 10% drop on the stability of its original system prompt.

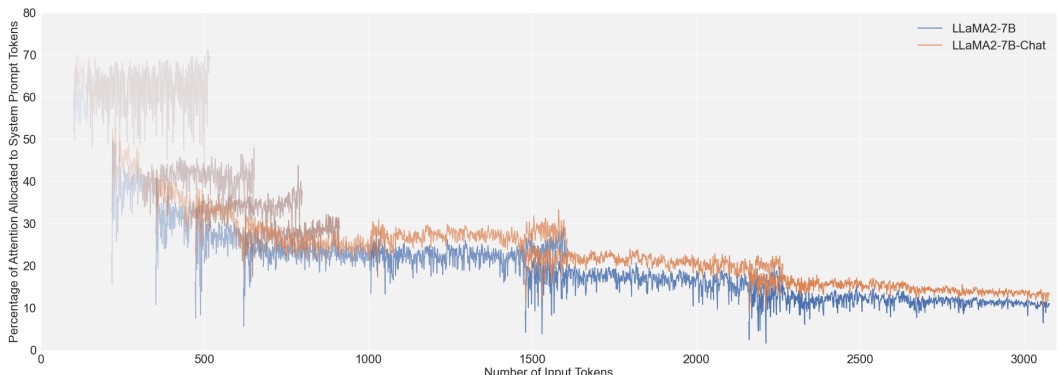

Figure 8: Comparison of attention decay between LLaMA2-7B before and after RLHF training. Different from the categorical palette used in Figure 4 to differentiate number of rounds when the answer is generated. The deeper the color, the later the round in which the answer is generated.

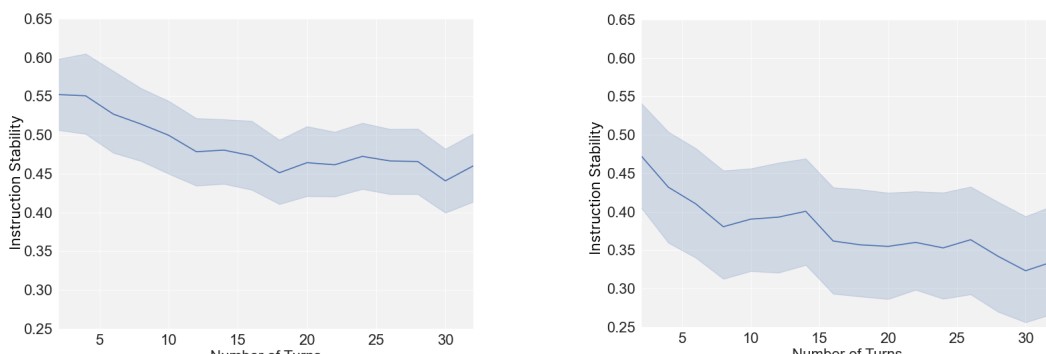

Figure 9: Measuring the instruction stability of `gpt-3.5-turbo-16k` via API using the same protocol as Figure 3. On the left, the system prompt is given to the API via the "system" argument; on the right, it is prepended to the user's first utterance.

## E    Discussion of Split-softmax Formula

We first quickly show how the post-intervention attention values in Equation 6 still form a distribution by summing up to 1, dropping subscript $t$:

$$
\begin{aligned}
\sum \alpha'_i &= \sum_{i \le |s_B|} \alpha'_i + \sum_{i > |s_B|} \alpha'_i \\
&= \frac{\pi^k(t)}{\pi(t)} \sum_{i \le |s_B|} \alpha_i + \frac{1 - \pi^k(t)}{1 - \pi(t)} \sum_{i > |s_B|} \alpha_i \\
&= \pi^k(t) + \left(1 - \pi^k(t)\right) \\
&= 1
\end{aligned}
$$

Meanwhile, it is worth-noting that the ratios of attention scores for tokens within the system prompt and within conversation history remain unchanged, thereby minimizing disruption to the attention mechanism.

