# OpenReview forum: "Measuring and Controlling Instruction (In)Stability in Language Model Dialogs"
_colmweb.org/COLM/2024/Conference — COLM_

### Official Review · Reviewer_Cp3m · 2024-05-06

**Rating:** 8
**Confidence:** 4
**Ethics Flag:** 1

**Summary:**

The paper proposes a method, namely split-softmax,  for avoiding the instruction drift present in conversations as the context increases. First, authors evaluate the instruction drift, through five categories: multi-choice response, answer character, answer-string format pattern, memorization of certain facts and languages the agent speaks. They simulated 200 conversations by prompting large language models (LLMs) for playing the user and  the system. They experimented with GPT3.5-turbo and LLaMA2-Chat-70B They implemented a function in python to compute the instruction stability.  Second, they observed stability decay in as the number of turns increase. Third, authors hypothesize this is due to the attention decay and they propose the split-softmax method to mitigate the instruction drift: a power-law scaling of attention.  They compare split-softmax against two baselines: prompt-repetition and classifier-free guidance. They show split-softmax is more stable as the number of turns increase than classifier-free. However, it has similar performance than  prompt-repetition although it requires less context.

**Questions To Authors:**

1) How was the instruction stability function  implemented per each of the five categories: multi-choice response, answer character, answer-string format pattern, memorization of certain facts and languages the agent speaks? Which libraries were used for each case?
2) Is the LLMs who plays the user more unstable that the one who plays the system according to Figure 3?
3) Fig 4. Is the behaviour of all the heads similar to the 11th head?
I suggest to move the Theorem A1 to the main paper.

Missing citations:
How Contextual are Contextualized Word Representations? Comparing the Geometry of BERT, ELMo, and GPT-2 Embeddings
Kawin Ethayarajh.
A Geometric Analysis of Neural Collapse with Unconstrained Features
Part of Advances in Neural Information Processing Systems 34 (NeurIPS 2021)

**Reasons To Accept:**

The paper clearly explains the instruction drift problem and it's cause. Authors design a way to evaluate instruction drift on 5 types of instructions: multi-choice response, answer character, answer-string format pattern, memorization of certain facts and languages the agent speaks. The do show a decrease of stability as the conversation gets longer.  Authors hypothesize this is due to the attention decay and they propose the split-softmax method to mitigate the instruction drift. They show split-softmax is more stable as the number of turns increase than classifier-free guidance. However, it has similar performance than  prompt-repetition although it requires less context.

**Reasons To Reject:**

It is not clear how the instruction stability function was implemented for each of the five categories: multi-choice response, answer character, answer-string format pattern, memorization of certain facts and languages the agent speaks. Authors limited to say that the function should return a number between 0 and 1 (however it is computed deterministically).
Fig 4, It would have been interesting to depict the stability decay in all the attention heads or in average.
Fig 6 shows that split-softmax is not better than system prompt repetition when the dialogue length increases, however it requires less  context.

---

> ### Author Rebuttal · Authors · 2024-05-29
>
> Thank you for the thoughtful comments and review!
>
> # Weakness
>
> **Details of stability functions.** The stability functions are diverse, but we make sure that all can be achieved with basic python functions deterministically. For example, below are some examples of the stability functions.
>
> ```python
> from nltk.sentiment import SentimentIntensityAnalyzer
> def get_sentiment(text, sentiment):
>
>     sia = SentimentIntensityAnalyzer()
>     score = sia.polarity_scores(text)[sentiment]
>     return score
> ```
>
> ```python
> lambda x: "tennis" in x.lower()
> ```
>
> ```python
> from langdetect import detect_langs
> def get_french_percentage(sentence):
>     languages_detected = detect_langs(sentence)
>     for lang in languages_detected:
>         if lang.lang == 'fr':
>             return lang.prob  # lang.prob is the probability of the detected language
>     return 0  # Return 0 if French was not detected
>
> ```
>
> **Figure 4 over different attention heads.** Preliminary experiments show that all attention heads, except those from the first several layers, demonstrate a similar attention decay pattern, as shown in Fig 4. We will add other specific heads or an average over all heads in the appendix.
>
> # Questions
>
> **Figure 3** didn’t express that the user LM could follow the instructions better. In fact, it should show a similar decay as the blue curve in Figure 3A since the roles of the two LMs are largely symmetric. The orange curve examines the agent LM’s behavior using the stability function of the user LM’s persona, therefore demonstrating that the agent LM is slowly picking up the persona of the user LM.
> **Theorem A1.** If accepted, we will consider moving more content from Appendix A into the main paper, given that the page limit is extended to 10.
> **Missing citations.** Thanks for pointing out relevant literature that we missed out. We have added them.

---

> > ### Comment · Reviewer_Cp3m · 2024-06-05
> > **Thank you for your answers**
> >
> > Thank you for addressing all my concerns and for taking them into account to further enhance your paper.

---

### Official Review · Reviewer_b2sN · 2024-05-12

**Rating:** 7
**Confidence:** 4
**Ethics Flag:** 1

**Summary:**

This paper presents a simple approach to addressing instruction drift in language model dialogs, particularly focusing on system-prompted chatbots. The authors introduce a quantitative benchmark for measuring instruction stability and propose a novel method, split-softmax, to mitigate attention decay, which they identify as a key factor contributing to instruction drift. The paper demonstrates through empirical testing on models like LLaMA2-chat-70B and GPT-3.5 that attention to initial system prompts decays over the course of long dialog exchanges. The split-softmax method, which requires no additional training or parameter adjustments, effectively enhances the model's focus on initial prompts, thereby improving instruction adherence over extended interactions.

**Questions To Authors:**

- Could you elaborate on how the split-softmax method might affect other aspects of model performance, such as response diversity or susceptibility to adversarial attacks?
- In extended dialog scenarios beyond the tested lengths, how does the split-softmax method perform in maintaining instruction stability?

**Reasons To Accept:**

- The paper introduces an original method for quantifying and addressing instruction drift in chatbots, which is a significant contribution to the field of dialog systems and AI safety.
- The empirical results are compelling, showing clear evidence of attention decay and its impact on instruction drift, along with a detailed comparison of the split-softmax method against strong baselines.
- Addressing instruction drift is crucial for the safe deployment of AI systems in real-world applications, making this work highly relevant and timely.
- I like this paper for its intuitive understanding of the attention decay phenomenon and backs this intuition with solid theoretical explanations.

**Reasons To Reject:**

- The discussion could benefit from a broader analysis of how these findings might influence other aspects of language model behavior and deployment in more varied contexts.
- While the paper compares the proposed method against strong baselines, including more diverse techniques could strengthen the claims.

---

> ### Author Rebuttal · Authors · 2024-05-29
>
> Thank you for the thoughtful comments and review!
>
> # Weakness
> **Discuss implications.** Thanks for the suggestion to extend the discussion of how the findings in this paper impact other aspects of LLM behavior under various deployment scenarios. We will add such a discussion:
> > The instruction drift indicates that current LM systems still cannot behave coherently over long horizons, which is the basis of long-term planning, such as information seeking (Lin et al., 2023) and tool usage (Nakano et al., 2021). Fundamentally, it is caused by a mismatch between their training schemes (text continuation or single-round RLHF) and their deployment scenarios (open-ended dialog with users). The phenomenon of instruction drift suggests there is much to be done to bridge this gap toward more robust and coherent dialog systems.
>
> **More diverse baselines.** We acknowledge that a more diverse set of techniques could strengthen the argument for the effectiveness of the proposed split-softmax technique. Among them, as suggested by reviewer 9T1g, studying different data compositions of the RLHF data could be especially useful—for example, adding role-playing data points into RLHF dataset could potentially improve the instruction stability of the resultant model.
>
> # Questions
> **Other aspects of model performances.** It would be a good idea to measure the language model’s output diversity with and without split-softmax. A naive guess is that it shouldn’t influence the entropy of the next-token distribution. We will add a few sentences of discussion on how split-softmax would help the model fend off adversarial attacks. We believe that attacks based on both traditional input optimization like GCG (Zou et al, 2023) as well as prompting like PAIR (Chao et al, 2023) could work less effectively on split-softmax-guarded models.
> **Dialogs beyond the tested lengths.** As seen in Fig 6, after 16 turns of dialog, the LLM has lost most of its instruction-following capability, with or without various mitigation methods. In fact, at this point, LLaMA-2-70B cannot even speak coherent English, but we could reasonably expect a more advanced model to maintain its language into >16 turns of dialog.

---

### Official Review · Reviewer_9T1g · 2024-05-14

**Rating:** 8
**Confidence:** 4
**Ethics Flag:** 1

**Summary:**

* The paper proposes a benchmark to measure the ability of language models to follow the system instruction over the course of a conversation.
* Using LLama2-70B and gpt-3.5-turbo, the paper shows that language models often suffer from significant instruction drift within 8 rounds of conversation.
* This decline in performance can be attributed to attention decay over the course of the conversation - with each turn, the model starts paying less attention to the system prompt.
* The paper proposes a method called split-softmax to combat this issue and improve adherence to the system prompt - this method performs better than simple, but intuitive baselines like repeating the system prompt at regular intervals and using classifier-free guidance.

**Questions To Authors:**

* It seems like choice of what type of data is used to construct the alignment blends/what goes into SFT & RLHF would be very important to counter this kind of behavior. Would fine-tuning on role-playing data/alignment blends help improve instruction stability?
* For the graph in appendix C, what would it look like if the instruction was in the first user turn instead of the system prompt?
* How are the probing questions created? Are they designed to be adversarial in nature?

**Reasons To Accept:**

* Very relevant and important direction of research - implications are useful for all fine-grained instruction following tasks that a language model assistant would be expected to perform (function calling, tool usage, staying in character)
* Excellent empirical analysis of instruction instability in language models and identification of the reason behind why that might be the case.
* The proposed method - split soft max - is very intuitive and easy to implement in practice.

**Reasons To Reject:**

* Would like to see some measure of helpfulness like MT-Bench to evaluate model behavior, instead of relying only on benchmarks like MMLU to quantify the impact of the intervention.
* There are not many details on the benchmark dataset or how the system prompts are chosen in the paper - would be helpful if more details are added.

---

> ### Author Rebuttal · Authors · 2024-05-29
>
> Thank you for the thoughtful comments and review!
>
> # Weakness
> **Other measures of helpfulness like MT-bench.** We agree that using a more comprehensive benchmark to measure the helpfulness of LLM could be a better idea than using MMLU. However, MT-bench requires users’ votes to determine the helpfulness of the uploaded model, which makes it hard to scale and obtain results in a timely manner.
> **More details of the benchmark dataset.** We will add more examples to the paper to provide a better understanding of the dataset.
>
>
> # Questions
> **Alignment data blend.** We could expect data points with a stronger connection to system prompts in the alignment dataset to be conducive to better instruction stability of the fine-tuned model. Studying the relationship between instruction stability and data composition could be a fruitful future research question.
> **For Figure 4 and Figure 8**, when the system prompt is used as the first user’s utterance, it’s reasonable to guess that the curve will look similar to Figure 4/8 on LLama models. In fact, for the prompt formatting scheme by llama, the system prompt is guarded with special system-prompt tokens and prepended to the first user message. So the suggested change would be equivalent to removing two special tokens. For GPT-3.5 experiment, it’s shown in Figure 9, suggests a special internal mechanism behind the GPT-3.5 API that emphasizes system prompts.
> **On probing questions.** The probing questions aren’t necessarily adversarial. For some system prompts, we stipulate certain rules that any utterances from the agent LM should follow; therefore, the probing questions can be arbitrary. In those cases, we use “What do you do in London as a tourist?” as a default choice. We will incorporate these details into the paper like for weakness 2.

---

### Decision · Program_Chairs · 2024-07-10

**Decision:**

Accept

**Comment:**

All reviewers appreciated the submission, in particular, the clear exposition of the problem of drift, the empirical analysis, and the introduction of the simple split-softmax method. The also mention this is a useful resource. There was also interesting conversation during rebuttal, where all participated. As all reviewers were in agreement with strong positive scores, I'd recommend acceptance.

I would recommend incorporating details on the benchmark dataset and system prompts as suggested by 9T1g in the CR version. Authors could also move the content from the appendix to the main paper as they suggested in rebuttal to Cp3m.